

# Phosphate solubilizing rhizobacteria isolated from jujube *ziziphus lotus* plant stimulate wheat germination rate and seedlings growth

Nidal Fahsi[1,2], Ismail Mahdi[1,3], Abdelhalem Mesfioui[2], Latefa Biskri[4,5] and Abdelmounaaim Allaoui[4]

[1] Institute of Biological Sciences (ISSB-P), Mohammed VI Polytechnic University (UM6P), Benguerir, Morocco
[2] Laboratory of Biologie & Sante, Faculty of Sciences, Ibn Tofail University, Kenitra, Morocco
[3] Laboratory of Microbial Biotechnologies, Agrobiosciences and Environement (BioMAgE), Faculty of Sciences Semlalia, University Cadi Ayyad, Marrakesh, Morocco
[4] Molecular Microbiology laboratory, Coalition Center of Innovation and Prevention of Epidemies in Morocco (CIPEM), Mohammed VI Polytechnic University (UM6P), Ben Guerir, Morocco
[5] African Genome Center (AGC), Mohammed VI Polytechnic University (UM6P), Ben Guerir, Morocco

Corresponding author
Abdelmounaaim Allaoui,
abdelmounaaim.allaoui@um6p.ma

## ABSTRACT

Jujube plant (*Ziziphus lotus* (L.) Desf.) can survive in arid climates and tolerates both biotic and abiotic stresses. Here, we isolated, for the first time in Morocco, nine phosphate solubilizing bacteria strains from jujube rhizosphere, designated J10 to J13, J15, & J153 to J156. Genotypic identification based on 16S rDNA sequencing, revealed six strains that belong to *Pseudomonas* (J10, J12, J13, J15, J153 and J154), two to *Bacillus* (J11 and J156), and one to *Paenibacillus* J155. Siderophores were produced by all strains. Proteases activity was missing in *Pseudomonas* sp. J153 & J154, whereas cellulase was restricted only to *Pseudomonas* sp. J10, *Paenibacillus xylanexedens* J155 and *Bacillus cereus* J156. Indole-3- acetic acid and ammonia were also produced by all strains, with a maxima of 204.28 µg mL$^{-1}$ in *Bacillus megaterium* J11 and 0.33 µmol mL$^{-1}$ in *Pseudomonas* sp. J153, respectively. *Pseudomonas* sp. J10 and *B. cereus* J156 grew on plates containing 1,500 µg mL$^{-1}$ of nickel nitrate, while *Pseudomonas* sp. J153 withstood 1,500 µg mL$^{-1}$ of either copper sulfate or cadmium sulfate. Phenotypic analysis of the potential of the isolates to promote early plant growth showed that wheat seeds inoculated with either *P. moraviensis* J12 or *B. cereus* J156 remarkably increased germination rate and seedlings growth. Lastly, antibiotic resistance profiling revealed that except for *Pseudomonas sp.* J11 and *B. cereus* J156, remaining strains displayed resistance at least to one of tested antibiotics. Collectively, *Pseudomonas* sp. J10, *P. moraviensis* J12, *Pseudomonas* sp. J153 and *B. cereus* J156, represent potential biofertilizers suitable for soils that are poor in P, and/or heavy metals contaminated.

## INTRODUCTION

Phosphorus (P) is considered one of the most important elements in plant nutrition after nitrogen. It is an essential macronutrient to all major metabolic processes in plants growth e.g., photosynthesis, energy transfer, respiration, and signal transduction (*Khan et al., 2010*; *Rahman et al., 2017*). Phosphate solubilizing microorganisms including bacteria play an important role in enhancing soil fertility and plant growth (*Miransari & Mackenzie, 2010*). Therefore, it is paramount to explore management strategies which are considered as an environmentally friendly process and economically feasible procedure to improve crop production and maximize their yields in P-poor soils (*Zaidi et al., 2009*). Exploration of the biodiversity of rhizobacteria and the optimization/manipulation of microbial interactions in the rhizosphere represents an imperative step towards formulating more efficient microbial inoculants with high P-solubilizing ability (*Khan, Zaidi & Wani, 2007*). Although P is plentiful in soils in both organic and inorganic forms, it is in unavailable forms for root uptake (*Sharma et al., 2013*). Numerous soil microorganisms particularly those present in plant's rhizosphere can release the bound forms of P to a soluble form to increase its bioavailability to plants (*Narayanasamy, Ghosh & Sarkar, 1981*; *Dubey et al., 1997*; *Dave & Patel, 2003*). PSB (phosphate solubilizing bacteria) belong to plant growth promoting rhizobacteria (PGPR) and are capable of solubilizing inorganic P from a variety of sources, such as dicalcium phosphate, tricalcium phosphate, or rock phosphate (*Khan et al., 2010*). Rhizobacteria are considered to be the best-known beneficial plant-associated bacteria and the most valuable bio-inoculants as they showed promising performances under controlled conditions such as the production of phytohormones, siderophores, phosphate solubilization, and nitrogen fixation (*Pérez-Montaño et al., 2014*).

A number of strategies have been documented to improve phosphorus availability in soils including, agronomic practices, organic amendments, composting, arbuscular mycorrhizal fungi, P efficient cultivars, and phosphate solubilizing microbes (*Kunwar, Lamichhane & Gauchan, 2018*). This later strategy is one of the most promising, as it is more sustainable and considered to be eco-friendly (*Gyaneshwar et al., 2002*). In contrast to other strategies, microbes have multiples benefits to plants as they contribute directly and indirectly to the nutrition, biocontrol of pathogens, and mitigating abiotic stresses (*Kunwar, Lamichhane & Gauchan, 2018*). Strains belonging to the bacterial genera *Pseudomonas, Bacillus, Rhizobacterium, and Enterobacter* are known as potent P solubilizing microorganisms (*Whitelaw, 1999*).

Antibiotic resistance is a major concern whose emergence and spreading rates are increasing. It is one of the major problems in deploying bacterial-based biofertilizers (*Kang et al., 2017*). Antibiotic resistance genes (ARGs) harbored by PGPB can be an inborn or gained property. Intrinsic resistance may be due to the presence of multidrug efflux pumps. This is supported by the phylogenetic analysis of several genes involved in antibiotic resistance which could be due to an evolutionary pattern (*D'Costa et al., 2011*; *Van Goethem et al., 2018*). Acquired antibiotic resistance may also reflect the acquisition of new resistance genes from other organisms by horizontal gene transfer from bacteria. The resistance of PGPR to antibiotics is a double-edged sword. On the one hand, resistant bacteria can serve

either as markers to monitor bacteria survival in vitro or in vivo (*Kluepfel, 1993*; *Trivedi et al., 2004*), and to help them competing in native and open microbial niches (*Cray et al., 2013*). On the other hand, their application in soil as bio-inoculants may represent potential risks by transferring ARG to other bacteria colonizing the same environment (*Ramakrishna, Yadav & Li, 2019*). However, a potential source of ARGs carried by PGPR and derived biocontrol agents and/or bio-fertilizers is widely forgotten and ignored (*Kang et al., 2017*). Hence, the urgency of large-scale introduction of the beneficial bacteria into soils can aggravate the situation leading to the spread of ARGs in the environment. In addition to antibiotic resistance, heavy metal pollution is an environmental concern that can have harmful effects on human health when they are taken up in amounts that cannot be processed by the organism (*Chauhan & Solanki, 2015*). Many toxic elements such as $Cu^{2+}$, $Pb^{2+}$, $Zn^{2+}$, $Cd^{2+}$ and $As^{3+}$ are generally found in detectable quantities in wastewater (*Pescod, 1992*). Some of these elements are necessary for plant growth, but a high concentration of them in wastewater becomes an obstacle (*Pescod, 1992*), which leads to the remediation of contaminated environments *via* sustainable methods (*Pizarro-Tobías et al., 2015*). Several methods are already being used to clean up the environment from these types of contaminants. The use of microorganisms capable of adsorbing heavy metal ions for bioremediation in contaminated soil is considered as an eco-friendly method and do not produce secondary pollution. Certain PGPR can reduce soil toxicity around plant roots. These PGPR also protect plants from being affected by toxic heavy metals (*Gamalero & Glick, 2011*). Various free-living rhizospheric PGPR can be used in contaminated soils to alleviate lethal effects of heavy-metals (*Belimov et al., 2004*). *Bacillus* and *Pseudomonas* exhibit vital role in the bioremediation of heavy metals (*Khan & Ahmad, 2006*; *Niu et al., 2011*).

The jujube tree (*Ziziphus lotus* (L.) Desf.) is a deciduous shrub belonging to the Rhamnaceae family. It is present in Morocco, in several biotopes of arid and semi-arid regions. It reaches 2 to 6 m, with tightly branched stems and smaller flowers and fruits (*Wang et al., 2016*). It grows on all soils: limestone, siliceous, clayey, and sandy, without human intervention, and it supports small amounts of salt (*Ionesco & Sauvage, 1969*). This shrub also behaves as a weed in several crops, including winter and spring cereals, food legumes and orchards in certain regions of Morocco such as Chaouia, Haouz, Zear, Rhamna and the Middle Atlas (*N & B, 2012*). It is considered as a multipurpose plant for arid and semi-arid areas where salt stress, drought, soil erosion and desertification (*Chadha, 2001*). In addition, jujube plant is traditionally used as sedative, bronchitis, antidiarrhea, skin infections, antiurinary troubles agents, antifever, insomnia agents, and hypoglycemic activities. These benefits are due to its richness in vitamins, polyphenols, amino acids, cyclopeptide alkaloids, minerals, and polyunsaturated fatty acids (*Lahlou, El Mahi & Hamamouchi, 2002*; *Adzu et al., 2003*; *Koley et al., 2016*).

The characterization of jujube rhizospheric PSB and their effects on plant growth is poorly understood and have not been extensively studied yet. The omnipresence of PGPR microorganisms in nature especially in rhizospheres, and their exceptional ability to enhance plant growth led us to undertake the present study. We isolated, for the first time in Morocco, PSB from rhizospheric soil of jujube plants (*Ziziphus lotus* (L.) Desf.).
| Table 1 | Physicochemical properties of the studied soil samples. | | |
| --- | --- | --- | --- |
| **Parameters** | | | **Results** |
| Granulometry (%)(AFNOR NF X31-107) | Clay | | 21,00 |
| | Fine silt | | 21,30 |
| | Coarse silt | | 11,50 |
| | Fine sand | | 21,50 |
| | Coarse sand | | 22,60 |
| Exchangeable elements (mg/kg)(AFNOR NF X31-108) | $K_2O$ | | 749 |
| | MgO | | 431 |
| | CaO | | 8071 |
| | $Na_2O$ | | 208 |
| pH-$H_2O$ (NF ISO 10390) | | | 8,7 |
| Electrical conductivity mS/cm (NF ISO 11265) | | | 0,15 |
| Total limestone (%) (NF ISO 10693) | | | 2,50 |
| Organic matter (%) (NF ISO 14235) | | | 2,92 |
| Oslen Phosphorus (mg/kg) (NF ISO 11263) | | | 17 |

We subsequently identified their genotype, assessed their conventional PGPR properties and tolerance to heavy metals and resistance to antibiotics in vitro, and finally evaluated their effects on wheat seeds germination.

# MATERIALS & METHODS

## Sampling and bacterial isolation

The sampling site, located in the experimental farm of Mohammed VI Polytechnic University (UM6P), Benguerir-Morocco (32.219731E, −7.892268N), is characterized by a temperate continental monsoon climate. The annual rainfall is 290, six mm and mainly occurs from October to January. Eight samples of roots and rhizospheric soil of jujube were collected from a 5 to 25 cm depth, packed and labeled in sterile plastic bags, then transported immediately to the laboratory in cool boxes. Fractions from all soil samples were mixed and analyzed for some physicochemical properties at the Agricultural Innovation and Technology Transfer Center (AITTC) of UM6P. The results are shown in Table 1. Rhizospheric samples were serially diluted under aseptic conditions by suspending 1 g of rhizosphere soil in 9 mL of sterile deionized water. Next, 100 μL of serial dilutions were subsequently plated on Tryptic Soy Agar medium (TSA) (EMD Millipore, Berlin, Germany) and incubated at $28 \pm 2$ °C till the appearance of bacterial colonies. Screening of individual colonies was carried out by repeated streaking.

## Screening of phosphate solubilizing bacteria

All bacterial isolates were qualitatively screened for inorganic P solubilization by inoculating a single colony of each strain in National Botanical Research Institute's Phosphate growth medium (NBRIP) containing $10 \, \mathrm{g \, L^{-1}}$ glucose; $0.1 \, \mathrm{g \, L^{-1}}$ (NH4)$_2$ SO$_4$; $5 \, \mathrm{g \, L^{-1}}$ MgCl$_2$ 6H$_2$O; $0.2 \, \mathrm{g \, L^{-1}}$ KCl, $0.25 \, \mathrm{g \, L^{-1}}$ MgSO$_4$,7H$_2$O and finally $5 \, \mathrm{g \, L^{-1}}$ Ca$_3$(PO$_4$)$_2$ (TCP: insoluble tricalcium phosphate) as a sole source of phosphate (*Nautiyal, 1999*). The initial media pH

was adjusted to 7.00 before use. Each bacterium was incubated on NBRIP plate at 30 °C for 7 days and only colonies surrounded by clear halos were selected for further studies as potential P solubilizer candidates. PSB were subsequently sub-cultured in TSB (Tryptic Soy Broth) (Professional lab, Casablanca, Morocco) liquid media and cryopreserved at −80 °C until use.

## Quantification of phosphate solubilization by bacteria

Inorganic P-solubilizing activity was quantified using TCP (in a modified NBRIP liquid medium). Briefly, bacterial suspension (0.1 mL of $OD_{600nm=}$ 0,8) was inoculated in a 100 mL flask containing 50 mL of NBRIP broth in triplicate. Non-inoculated medium was used as blank, while *Rhizobium tropici* CIAT 899 served as a positive control. Bacterial cultures were incubated at $28 \pm 2$ °C during five days under shaking condition at 150 rpm. The cultures were then harvested by centrifugation at 13.000 rpm for 10 min and the soluble P, contained in the supernatant, was quantified by colorimetric method using SKALAR (SKALAR SAN++ SYSTEM). Dissolved P concentration was determined by subtracting the P concentration of the blank from the final concentration of soluble P in the inoculated broths. The final pH of each culture supernatant was also measured. The experiments were performed in triplicate and the results are means of the replicates.

## Bacterial antibiotic resistance and heavy metal tolerance

Antibiotic resistance profile of selected PSB was determined using TSA medium supplemented with selected antibiotics namely kanamycin (50 $\mu$g mL$^{-1}$), streptomycin (100 $\mu$g mL$^{-1}$), tetracycline (10 $\mu$g mL$^{-1}$), ampicillin (100 $\mu$g mL$^{-1}$), chloramphenicol (20 $\mu$g mL$^{-1}$) and spectinomycin (60 $\mu$g mL$^{-1}$).

Heavy metal tolerance of selected isolates was tested using the same method (TSA plates) with the addition of increasing concentrations (ranging from 0 to 1500 $\mu$g mL$^{-1}$) of three heavy metals; cadmium sulfate ($CdSO_4$), copper sulfate ($CuSO_4.5H_2O$) and nickel nitrate ($N_2NiO_6$). The plates were incubated at 30 °C for 24 h.

## Strains genotyping using 16S rRNA gene sequencing

PSB identification was performed using 16S rRNA gene sequencing. The polymerase chain reactions (PCR) were carried out directly with fresh bacterial suspension, using a pair of universal primers pA (5′- AGAGTTTGATCCTGGCTCAG-3′) and 926R_Quince (degenerated one) (5′-CCG YCAATTYMTTTRAGTTT-3′), and MyTaq Mix, 2X (ThermoFisher, Casablanca, Morocco) containing Taq DNA polymerase, dNTP, MgCl$_2$ and buffer. Amplification of 16S rDNA sequences was made in 50 $\mu$l reaction mixture containing 25 $\mu$L of MyTaq mix, 1 $\mu$L of each primer (20 $\mu$M), 22 $\mu$L of DNase/RNase-free distilled water and 1 $\mu$L of overnight bacterial culture as DNA template. The reaction was performed in a VWR® thermal cycler using the following PCR optimized conditions: initial denaturation at 95 °C for 5 min, 35 cycles of denaturation at 95 °C for 30 s, annealing at 52 °C for 30 s, elongation at 72 °C for 1 min, and final elongation at 72 °C during 10 min. The amplified 16S rDNA fragments (910-bp) were sequenced by Genome Quebec, Canada. The generated DNA sequences were aligned to available standard sequences of bacterial lineage in the National Center for Biotechnology Information GenBank database

(http://www.ncbi.nlm.nih.gov/) and the High-Quality Ribosomal RNA databases SILVA (https://www.arb-silva.de) using BLAST algorithm to carry out a taxonomic assignment of each isolate. The phylogenetic tree of identified PSB was built using Ugene software.

### Indole-3- acetic acid (IAA) measurement assay

Bacteria were analyzed for the quantitative determination of indole acetic acid (IAA) production. For this purpose, 100 µL of each PSB strain ($OD_{600nm} = 0.8$) was grown in 50 mL Tryptic Soy Broth (TSB) supplemented with 0.1% L-tryptophan as IAA precursor at $28 \pm 2 °C$ in a shaking incubator at 200 rpm. After 7 days, two mL of Salkowski reagent [0.5M $FeCl_3$: 70% perchloric acid/water (2:49:49 ratio)] (*Glickmann & Dessaux, 1995*), was pipetted into test tubes containing one mL of culture supernatant filtrates. The tubes containing the mixture were gently vortexed and left for 30 min in dark for the development of color at room temperature ($26 \pm 2 °C$). The absorbance was determined at an $OD_{535nm}$. The quantity of IAA produced in each supernatant was estimated in ($\mu g\ mL^{-1}$) from a calibration curve using a standard IAA (Sigma Aldrich, Overijse, Belgium).

### Siderophores production assay

Qualitative production of siderophores by selected strains was detected on the chrome-azurol S (CAS) medium as previously described (*Schwyn & Neilands, 1987*). Briefly, each bacterial culture was spot-inoculated separately on CAS agar plates. The plates were kept at 30 °C for 3 days. After the incubation period, the appearance of orange halo (blue to yellow/orange) around the colony was considered as a positive result for siderophores production.

### Extracellular enzymes production assay

Bacteria were qualitatively analyzed for the production of protease and cellulase by the plate method (*Kavitha, Nelson & Jesi, 2013*). Protease activity (casein degradation) was tested by inoculation of selected strains into nutrient agar medium containing casein 5 g $L^{-1}$, yeast extract 2.5 g $L^{-1}$, glucose 1 g $L^{-1}$, and agar 15 g $L^{-1}$ and amended with 10% of skim milk. After 48 h incubation at 30 °C, a clear zone around colonies indicated positive proteolytic activity. For cellulase activity, a mineral–salt agar plate containing 0.4% $(NH_4)_2SO_4$, 0.6% NaCl, 0.1% $K_2HPO_4$, 0.01% $MgSO_4$, 0.01% $CaCl_2$ with 0.5% carboxymethyl cellulose, and 2% agar were surface-inoculated with each strain and incubated 48 h at 30 °C. Plates were stained with 0.1% Congo Red (Sigma Aldrich, Casablanca, Morocco) for 15 min. Following de-staining during 15 min, using 1 M NaCl, the development of the halo zone around the colonies reflects cellulase production.

### Ammonia production assay

Bacteria strains were tested, qualitatively and quantitatively, for ammonia production in peptone water as previously described (*Cappuccino & Sherman, 1992*). Briefly, freshly grown cultures were inoculated into 10 mL peptone water and incubated for 48 h at 30 °C on a shaker (150 rpm). Post incubation period, 0.5 mL of Nessler's reagent was added to each tube. Ammonia production is proportional to the brown color intensity. It was measured spectrophotometrically at $OD_{450nm}$ using the VICTOR NivoTM Multimode

Plate Reader (PerkinElmer, Casablanca, Morocco) and determined using a standard curve prepared with 0.1–1 $\mu$mol mL$^{-1}$ ammonium sulfate.

## Wheat seeds germination assay

Our selected strains were assessed for their effect on seed germination. Seeds of durum wheat (Variety vitron) were surface sterilized with 2% sodium hypochlorite solution for 1 min, rinsed thoroughly with sterile distilled water, soaked in 70% ethanol for 1 min and washed 5 times in single distilled water followed by air-drying. PSB cell pellets were obtained by centrifuging an overnight culture (OD$_{600nm}$ =0,8) at 10. 000 rpm for 5 min, the supernatant was removed, and the pellets were resuspended in 5 mL of sterile distilled water, vortexed and used for seed treatment. Fifteen sterilized seeds were treated with 5 mL of bacterial suspension for 30 min, air-dried, and then placed on sterile Petri dishes containing 0.7% agar medium and incubated at 25 °C. Triplicates were maintained for each treatment. Seeds were surface sterilized with 2% sodium hypochlorite solution for 1 min, rinsed thoroughly with sterile distilled water. Next, seeds were incubated in a dark incubator for 48 h, then left at room temperature in a day/night cycle. The germination rate was recorded after 24 h and 48 h. Root length, shoot length, fresh weight, and dry weight were measured after 7 days. The germination rate and vigor index were calculated using the following formula (*Islam et al., 2016*):

$$Germination\ rate\,(\%) = \frac{Number\ of\ seeds\ germinated}{Total\ number\ of\ seeds} \times 100\,(1) \tag{1}$$

$$Vigor\ index = (\%)\,Germination \times Total\ plant\ lenght. \tag{2}$$

## Statistical analysis

Results presented here are the mean of triplicates ($n = 3$) $\pm$ Standard deviation. Statistical analysis was performed using IBM SPSS statistics 20 for windows. The differences between treatments were statistically analyzed using analysis of variance (ANOVA) and subsequently by Tukey's multiple range test at $p < 0.05$.

# RESULTS

## Bacteria screening identified nine best phosphate solubilizing strains

The screening of P solubilizing bacteria from different rhizospheric soil samples of jujube on NBRIP led to the isolation of forty-one bacterial isolates. This microbial population has different aspects, but all exhibited a common character of tricalcium phosphate (Ca$_3$(PO$_4$)$_2$) solubilization on solid medium. Indeed, bacterial isolates were able to form a clear zone (halo) around their colonies on the NBRIP medium, indicating positive solubilization of P from tricalcium phosphate (TCP). Nine isolates were selected as being the best performers on plates and named J10 to J13, J15, and J153 to J156 (J for Jujube). Next, we tested their ability to solubilize inorganic phosphorus (Ca$_3$(PO$_4$)$_2$) in NBRIP liquid medium. The amount of soluble P and growth media's pH were measured 5 days

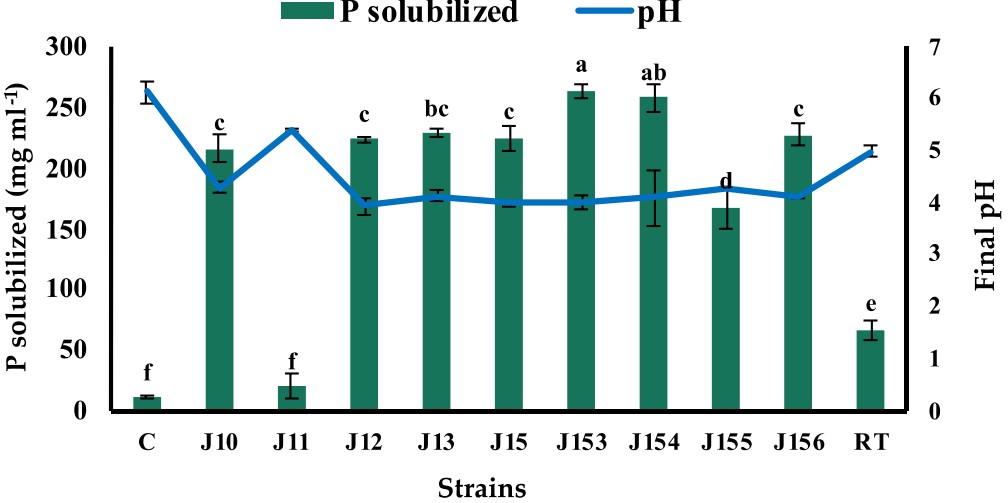

**Figure 1 Phosphate solubilization by isolated strains.** Concentrations of solubilized P released in NBRIP broth of isolated PSB strains and the resulting pH of culture supernatants indicated by the blue line. C (Negative control: non-inoculated medium), RT (*Rhizobium tropici*: positive control). The values represent means of 3 replicates ($n = 3$) $\pm$ standard deviations. Different letters indicate significant differences at $p < 0.05$.

post incubation. Eight strains were found to release P from TCP with concentrations ranging from 20.5 mg L$^{-1}$ to 264 mg L$^{-1}$ (Fig. 1). Remarkably, the highest solubilization was recorded for strain J153, while the lowest one (20.5 mg L$^{-1}$) was measured for strain J11. The amount of P solubilization by the referenced strain, *Rhizobium tropici* did not exceed 67.5 mg L$^{-1}$. As expected, P solubilization was accompanied by a significant drop in pH, of the culture media, from 7.0 to 4.0 (Fig. 1).

## Strains J10-13, J15 and J153-156 belong to the genera of *Pseudomonas*, *Bacillus*, and *Paenibacillus*

Characterization of the nine PBS strains to the genus level was performed by 16S ribosomal DNA gene partial sequencing. Generated sequences of 900-bp length were aligned to available 16S rDNA sequences using GenBank and SILVA databases. As summarized in Table 2, three strains (J12, J13, and J15) show 98% identity to the 16S rRNA gene sequences of *Pseudomonas moraviensis*; three strains J10, J153, and J154 share 98–99% identity to *Pseudomonas* sp.; two strains J11 and J156 exhibit 98 and 99% identity to *Bacillus megaterium* and *Bacillus cereus*, respectively. Lastly strain J155 shares 98% identity to *Paenibacillus xylanexedens* (Fig. 2).

## Except for *Pseudomonas* sp. J11 and *B. cereus* J156, remaining strains displayed resistance at least to one antibiotic

PGPB tend to harbor genes that confer resistance to antibiotics (*Kang et al., 2017*). To assess bacterial resistance to antibiotics, we checked our strains for growth on plates supplemented with a set of different antibiotics, frequently encountered among bacteria isolated from soils. As reported in Table 3, out of the nine tested strains, seven presented
**Table 2 Molecular identification of selected PSB using 16S rRNA gene sequencing.**

| Isolate | Closest species | Identity (%) | Accession No. |
|---------|-----------------|--------------|---------------|
| J10 | *Pseudomonas* sp. | 98 | MT771625 |
| J11 | *Bacillus megaterium* | 98 | MT771626 |
| J12 | *Pseudomonas moraviensis* | 98 | MT771627 |
| J13 | *Pseudomonas moraviensis* | 98 | MT771628 |
| J15 | *Pseudomonas moraviensis* | 98 | MT771629 |
| J153 | *Pseudomonas* sp. | 99 | MT771630 |
| J154 | *Pseudomonas* sp. | 98 | MT771631 |
| J155 | *Paenibacillus xylanexedens* | 98 | MT771632 |
| J156 | *Bacillus cereus* | 99 | MT771633 |

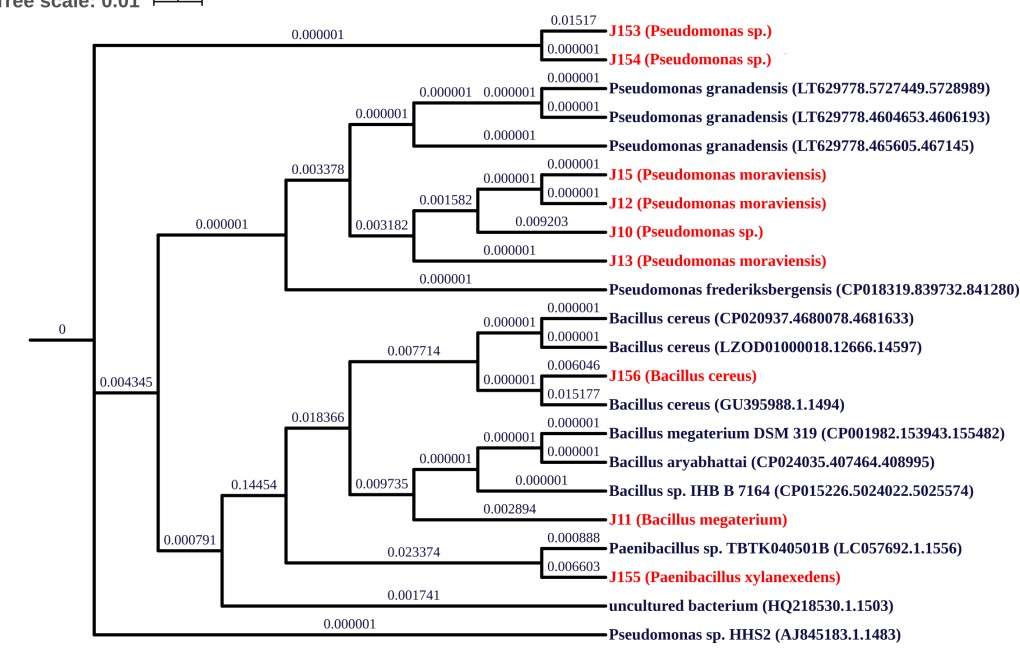

**Figure 2 Phylogenetic tree of isolated strains.** Neighbor-joining phylogenetic tree showing relationship between the selected PSB from jujube soil and their representative species from NCBI database, built using UGENE Software. Multiple alignment of 16S rRNA gene sequences was performed using NCBI database.

resistance at least to one antibiotic. Strains *P. moraviensis* J12 & J15, *Pseudomonas* sp. J153 & J154 resist to chloramphenicol and ampicillin, while *Pseudomonas* sp. J10 & J13 confer resistance to chloramphenicol, ampicillin, and spectinomycin. Strains *Pseudomonas* sp. J11 and *Bacillus cereus* J156 are sensitive to all tested antibiotics. Strain *P. xylanexedens* J155 is resistant to both kanamycin and spectinomycin. Lastly, none of the tested strains are resistant to neither streptomycin nor tetracycline (Table 3). In the next steps, to avoid any potential contamination, we took advantage to these resistances to grow bacteria on selective media.

**Table 3  Summary table of relevant phenotypic traits observed in selected PSB using plate assay.**

| Strain | Antibiotic resistance | Siderophores production | Extracellular enzymes | |
|---|---|---|---|---|
| | | | Proteases | Cellulase |
| *Pseudomonas* sp. J10 | Amp$^R$, Cm$^R$, Spect$^R$ | ++ | + | + |
| *B. megaterium* J11 | – | + | + | – |
| *P. moraviensis* J12 | Amp$^R$, Cm$^R$ | ++ | + | – |
| *P. moraviensis* J13 | Amp$^R$, Cm$^R$, Spect$^R$ | +++ | + | – |
| *P. moraviensis* J15 | Amp$^R$, Cm$^R$ | ++ | + | – |
| *Pseudomonas* sp. J153 | Amp$^R$, Cm$^R$ | ++ | – | – |
| *Pseudomonas* sp. J154 | Amp$^R$, Cm$^R$ | ++ | – | – |
| *Paenibacillus xylanexedens* J155 | Kan$^R$, Spect$^R$ | ++ | + | + |
| *B. cereus* J156 | – | + | + | + |

### *Pseudomonas* sp. J153 and *B. cereus* J156 withstand high concentrations of copper sulfate/cadmium sulfate and copper sulfate/nickel nitrate, respectively

Heavy metals such as lead (Pb), cadmium (Cd), chromium (Cr), and mercury (Hg) are exceptionally toxic and dangerous environmental pollutants (*Tangahu et al., 2011*). We investigated the capacity of our strains to grow under various concentrations of nickel nitrate ($N_2NiO_6$), copper sulfate pentahydrate ($CuSO_4.5H_2O$) and cadmium sulfate ($CdSO_4$). The nine strains exhibited various tolerance characteristics (Table 4). In the copper assay, *Pseudomonas* sp. J153 grows up to 1,500 µg mL$^{-1}$, *Bacillus cereus* J156 to 1,000 µg mL$^{-1}$, while strains *Pseudomonas* sp. J10, *P. moraviensis* J12, *P. moraviensis* J13, and *P. moraviensis* J15 supported a maximum of 500 µg mL$^{-1}$. The lowest tolerated concentration, 300 and 200 µg mL$^{-1}$ were seen in *B. megaterium* J11 and *Pseudomonas* sp. J154 strains, respectively. When tested for cadmium sulfate, only *Pseudomonas* sp. J153 grows up to 1500 µg mL$^{-1}$, whereas 300 µg mL$^{-1}$ was the maximal concentration tolerated by strains *Pseudomonas* sp. J10, *P. moraviensis* J12, *P. moraviensis* J13, *P. moraviensis* J15, *Pseudomonas* sp. J154 and *B. cereus* J156. Lastly, low tolerance at 100 µg mL$^{-1}$ and 10 µg mL$^{-1}$ were detected in strains *P. xylanexedens* J155 and *B. megaterium* J11, respectively (Table 4). In the nickel nitrate assay, *Pseudomonas* sp. J10 and *Bacillus cereus* J156 strains, grow up to 1,500 µg mL$^{-1}$ and strain *P. xylanexedens* J155 tolerated the lowest concentration of 300 µg mL$^{-1}$. However, the remaining strains tolerate growth up to 500 µg mL$^{-1}$. Taking together, our data highlighted the remarkable capacity of strains *Pseudomonas* sp. J153 and *B. cereus* J156 to withstand abnormal high concentrations of both copper sulfate/cadmium sulfate and copper sulfate/nickel nitrate, respectively.

### *B. megaterium* J11 is the best indole-3-acetic acid producer

The production of indole acetic acid (IAA) is a major property shared by numerous rhizospheric bacteria that stimulate plant growth (*Mohite, 2013*). Seven days post incubation, all tested strains produced IAA although at various levels, ranging from 57.1 to 204.28 µg mL$^{-1}$ (Fig. 3). The highest concentration was produced by *B. megaterium* J11, whereas the lowest one was measured for *P. xylanexedens* J155 (Fig. 3).

**Table 4** Heavy metals tolerance of selected PSB strains.

| Strain | $CuSO_4 \cdot 5H_2O$ (mg L-1) | | | | $CdSO_4$ (mg L-1) | | | | $N_2NiO_6$ (mg L$^{-1}$) | | | |
|---|---|---|---|---|---|---|---|---|---|---|---|---|
| | 0.3 | 0.5 | 1 | 1.5 | 0.3 | 0.5 | 1 | 1.5 | 0.3 | 0.5 | 1 | 1.5 |
| *Pseudomonas* sp. J10 | + | + | − | − | + | − | − | − | + | + | + | + |
| *B. megaterium* J11 | + | − | − | − | - | − | − | − | + | + | − | − |
| *P. moraviensis* J12 | + | + | − | − | + | − | − | − | + | + | − | − |
| *P. moraviensis* J13 | + | + | − | − | + | − | − | − | + | + | − | − |
| *Pseudomonas* sp. J153 | + | + | + | + | + | + | + | + | + | + | − | − |
| *Pseudomonas* sp. J154 | − | − | − | − | + | − | − | − | + | + | − | − |
| *Paenibacillus xylanexedens* J155 | + | − | − | − | - | − | − | − | + | − | − | − |
| *B. cereus* J156 | + | + | + | − | + | − | − | − | + | + | + | + |
| *E. coli* DH5 α | + | + | + | − | + | − | − | − | + | + | − | − |

**Notes.**
+, for tolerance; −, for sensitivity.

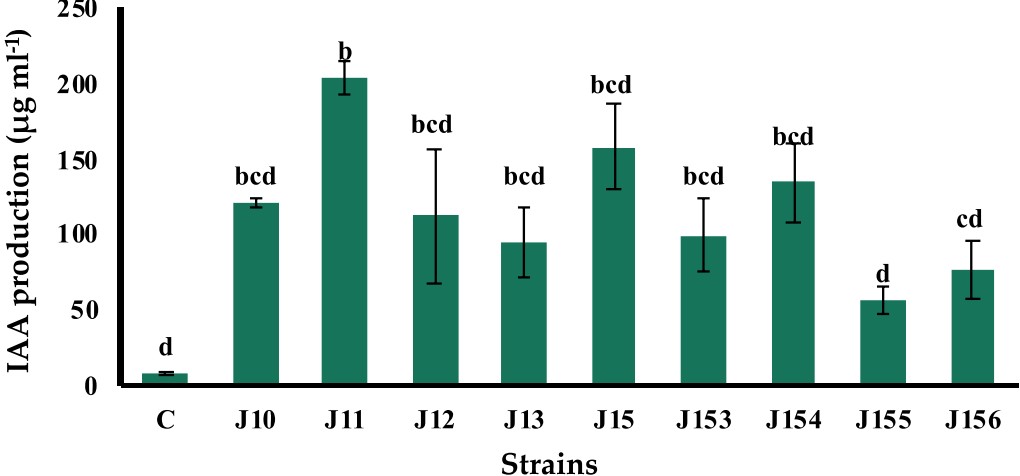

**Figure 3 Indole acetic acid production by isolated strains.** Indole acetic acid production by selected PSB in the TSB broth amended with 0,1% of L-tryptophan. C (Negative control: non-inoculated medium). The values represent means of three replicates ($n = 3$) ± standard deviations. Different letters indicate significant differences at $p < 0.05$.

### *P. moraviensis* J13 is the best siderophores producer

Siderophores are best known for binding iron (Fe) and for mobilizing soil-immobilized Fe, although they may also contribute to improve the availability of P to plants (*Sharma et al., 2013*), by solubilizing minerals and chelating heavy metals, which in turn increases nutrient uptake and plant growth (*Gontia-Mishra et al., 2016*). The ability, in vitro, of selected PSB, to produce siderophores was qualitatively estimated using the CAS-agar plate assay. All tested strains were able to produce siderophores, although at various levels as deduced by the size of the halo zone and the intensity of the color change of the CAS-Agar (Table 3). *P. moraviensis* J13 was the most efficient siderophores producer, the six strains (*Pseudomonas* sp. J10, J153 & J154, *P. moraviensis* J12 & J15, and *P. xylanexedens* J155

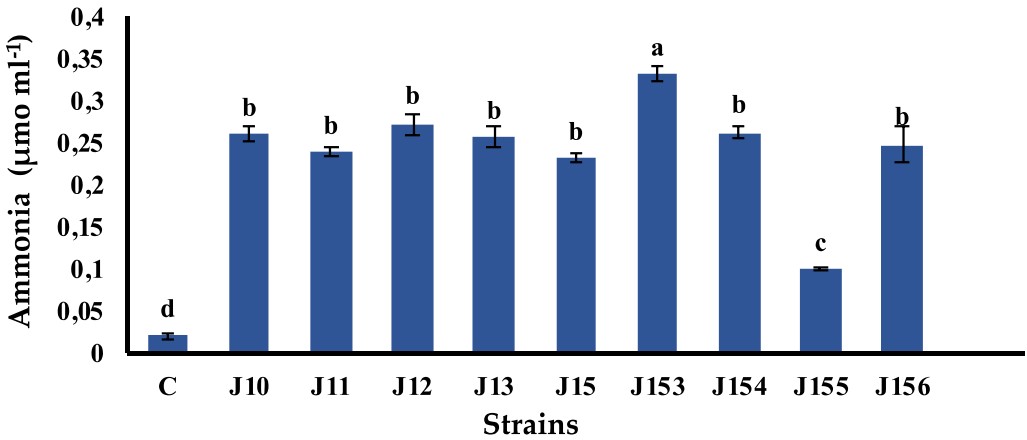

**Figure 4 Ammonia production by isolated strains.** Ammonia production by selected PSB. C (Negative control: non-inoculated medium). The values represent means of three replicates ($n = 3$) ± standard deviations. Different letters indicate significant differences at $p < 0.05$.

produced intermediate level, whereas the lowest production was seen in both *B. megaterium* J11 and *B. cereus* J156 strains.

### *Pseudomonas* sp. J153 is the best ammonia producer

Ammonia is a chemical compound having indirect plant health benefits, primarily by acting as metabolic inhibitor against phytopathogens (*Kumar et al., 2012*). All tested strains were able to produce ammonia with various concentrations. The highest value of 0.33 µmol mL$^{-1}$ was detected in *Pseudomonas* sp. J153, while the lowest one, 0.1 µmol mL$^{-1}$, was measured in *P. xylanexedens* J155 (Fig. 4).

### Proteases are not produced by *Pseudomonas* sp. J153 and J154 and cellulase activity is restricted to *Pseudomonas* sp. J10, J155 and *B. cereus* J156

Bacterial extracellular enzymes such as proteases and cellulases play a dual important role in the biological control of phytopathogens and in soil fertilization (*Mitchell & Alexander, 1963*). The nine strains were tested for their ability to produce proteases and cellulases. Results of both proteases and cellulase assay are shown in Table 3. As for proteases production, except for *Pseudomonas* sp. J153 & J154, the remaining seven strains developed halo zone around the colonies. As a control, no halo zone was seen using *E. coli* strain DH5 $\alpha$, used here as a negative control. Cellulase activity was solely detected in three strains: *Pseudomonas* sp. J10, *P. xylanexedens* J155, and *B. cereus* J156, each of which formed a yellow/whitish zone around their colonies and were considered as cellulase positive. No cellulase activity was observed in the remaining six other strains.

### Inoculation with *P. moraviensis* J12 and *B. cereus* J156 promote the highest rate of wheat seeds germination and seedlings growth

The treatment of wheat seeds by the nine PSB strains had a significant effect ($P < 0.05$) on the germination rate and wheat vigor index, as compared to the control

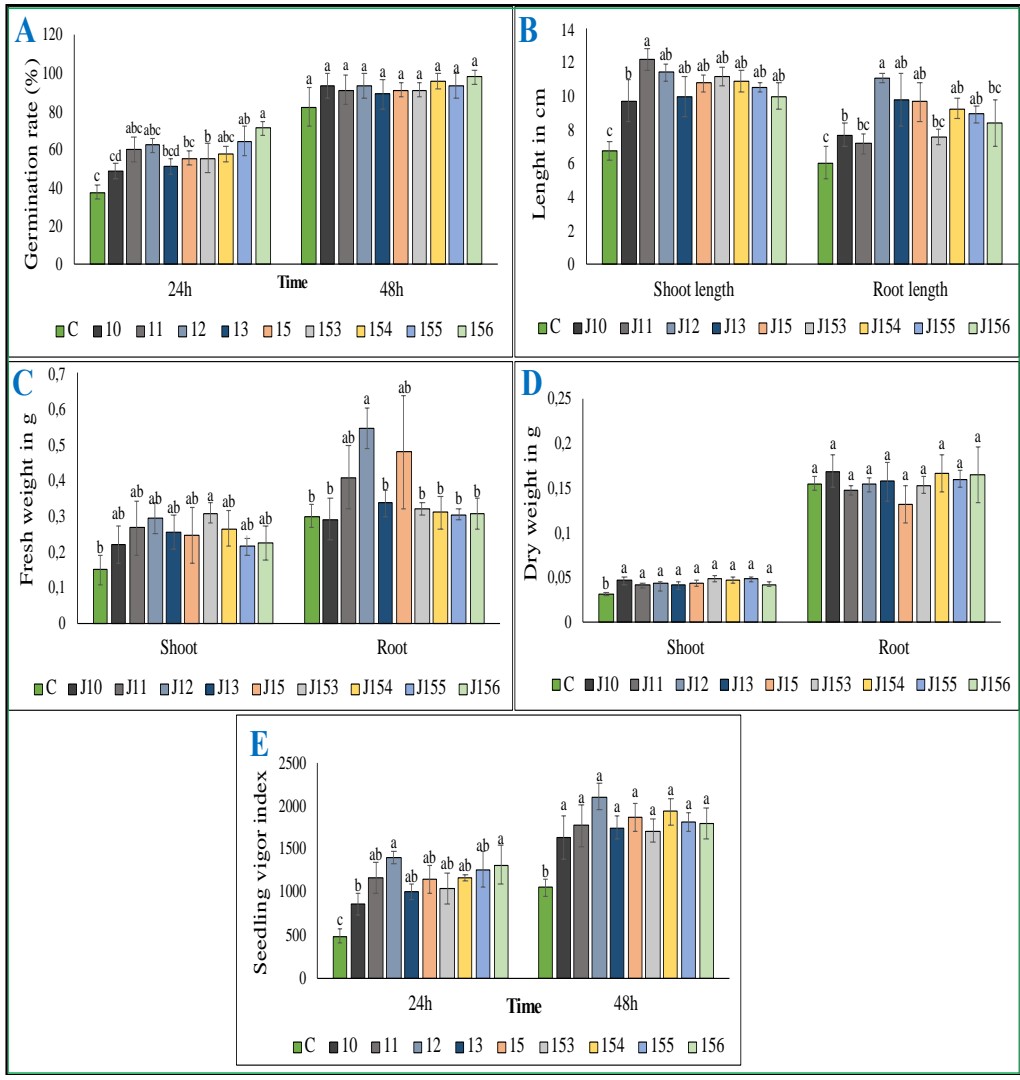

**Figure 5** **Effect of bacterial inoculation of wheat seed germination parameters.** Effect of bacterial inoculation of wheat seed germination parameters. (A) Germination rate after 24 and 48 h of incubation (B) Total length of shoots and roots after 7 days, (C) Fresh weight of shoots and roots after 7 days, (D) Dry weight of shoots and roots after 8 days, (E) Seedling vigor index. c (Negative control: no-inoculated seeds). Different letters indicate significant differences at $p < 0.05$.

(Figs. 5A et 5E). However, these effects varied depending on the PSB isolates. For instance, both *P. moraviensis* J12 and *B. cereus* J156 strains were the most efficient in promoting wheat germination as represented by vigor index (Fig. 5E). Results revealed that, compared to non-inoculated control, seeds inoculated by each of the nine strains showed a considerable impact on different growth parameters (Fig. 6).

Regarding shoot and root length after 7 days of growth, seeds inoculated with all strains, especially *B. megaterium* J11 significantly enhanced shoot and root length ($p < 0.05$). Maximum root length was seen upon inoculation with *P. moraviensis* J12 (Fig. 5B). We

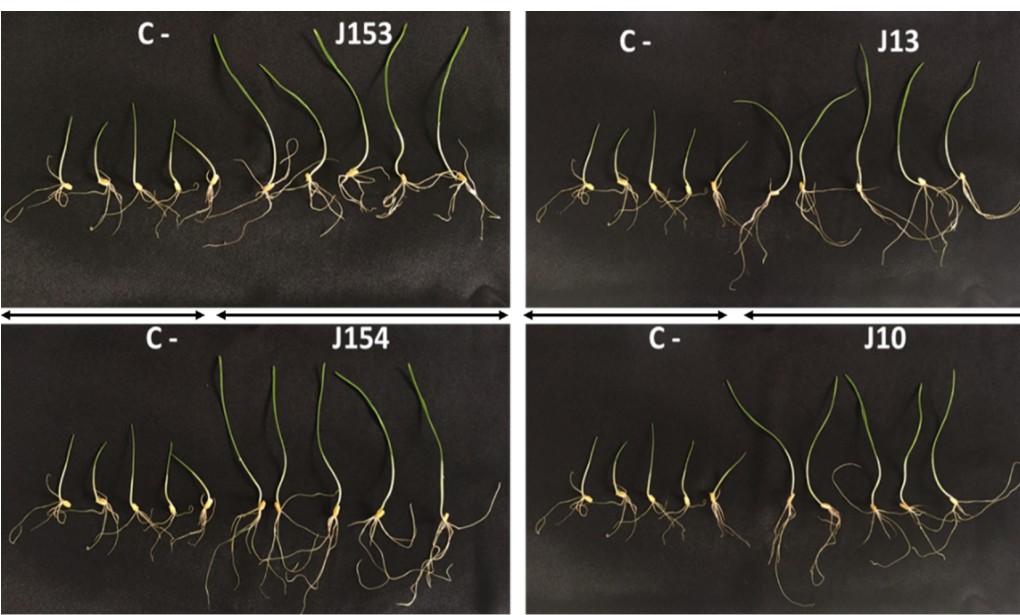

**Figure 6** **Effect of studied strains on shoots and roots growth.** Effect on shoots and roots growth after seven days of inoculation of wheat seeds by the four *pseudomonas* strains: *P. moraviensis* J13, *Pseudomonas* Sp. J10, J153 and J154 (C-:control non-iniculated seeds).

also noticed, whatever the nature of the inoculum was, seeds root dry weights remain unchanged. In contrast, a significant increase in shoot dry weight was detected (Fig. 5D). Furthermore, wheat seeds inoculation significantly affected shoot fresh weight, but not root dry weight except for *B. megaterium* J11, *P. moraviensis* J12, and J15 strains (Fig. 5C).

## DISCUSSION

Phosphorus is an important limiting factor in agriculture production and microbial P solubilization seems to be an effective process to release the precipitated P in soil. In the present work, we isolated and screened nine P solubilizing bacteria from Jujube roots. Genotyping analysis revealed that these strains belong to the genera of *Pseudomonas*, *Bacillus,* and *Paenibacillus*. A recent study based on ACC-deaminase-producing screening criteria, reported that *Pseudomonas* and *Bacillus* genus are among species that are associated with jujube rhizospheric plant with the dominance of *Pseudomonas* genus (*Zhang et al., 2020*). Our isolated bacteria were characterized as being P solubilizers with values ranging from 20.5 to 264 mg $L^{-1}$. Comparatively, *Zhang et al. (2020)*, reported that *Pseudomonas lini* (KM349410) isolated from jujube rhizosphere, was the best P solubilizing strain at 69 mg $L^{-1}$. We found here that the highest concentration of dissolved P in the medium was recorded by genus *Pseudomonas* followed by *Paenibacillus*, and then by our two *Bacillus* strains. Not surprisingly given, the pH of all bacterial cultures dropped significantly from 7.0 to 4.0, likely due of organic acids production (*Pandey & Maheshwari, 2007*; *Khan, Zaidi & Ahmad, 2014*; *Otieno et al., 2015*). Indeed, different genera of bacteria uses multiple P
solubilization mechanisms, in addition to the production of acids, such as chelation and siderophores (*Pandey & Maheshwari, 2007*).

Indole-3-acetic acid has been associated with plant growth promoting effect of several rhizospheric microorganisms that stimulate elongation and proliferation of root systems (*Glick, 2012*; *Shilev, 2013*; *Otieno et al., 2015*; *Thomas, Murphy & Murray, 2016*; *Mazumdar, Saha & Ghosh, 2019*). IAA is also frequently used as a marker to select beneficial bacteria (*Went & Thimann, 1937*). Auxin production stimulates root development resulting in a higher nutrients and water absorption from the soil (*Höflich, Wiehe & Kühn, 1994*). *B. megaterium* J11 is the best IAA producer (204.28 $\mu$g mL$^{-1}$), and the lowest in-term of P solubilization (20.5 mg L$^{-1}$), but significantly stimulates wheat seedling. Recent genome sequencing of *Bacillus megaterium* TRQ8 isolated from wheat (*Triticum turgidum* subsp. durum) rhizosphere, revealed the presence of genes specifying factors associated with bacteria–plant interactions, *i.e.*, auxin biosynthesis (indoles), phosphate metabolism, siderophores production, and osmotic/oxidative stress response (*Montoya, Cota & De los Santos Villalobos, 2019*).

Our isolated strains produced various levels of siderophores but *P. moraviensis* J13 was the best producer. During plant-bacteria association, siderophores production is beneficial to plant and are considered as an important trait of PGPR (*Bal et al., 2013*), that may influence plant growth as they mobilize different metal ions and play also an important role in bio-control (*Matthijs et al., 2007*). The *P. moraviensis* strain J12 exhibits a good production of siderophores in addition to be the highest P solubilizer, promotes shoot and root length, root fresh weight and stimulates wheat seeds germination. Compared to *P. moraviensis* J13, strain *P. moraviensis* J12 increases root fresh weight. The third strain of *P. moraviensis* J15 increases root and shoot length, shoot dry weight and stimulates wheat seedling.

Ammonia production is an essential PGPR trait often associated with plant growth (*Yadav, Verma & Tiwari, 2010*). We found that *Pseudomonas* sp. J153 is the best ammonia producer. Remarkably, both *Pseudomonas* sp. J153 and J154 strains are lacking both proteases and cellulase activities, usually required to degrade the cell walls of phytopathogens (*Hameeda et al., 2008*; *Nagpure, Choudhary & Gupta, 2014*). These activities are required during plant-microbe interactions and in roots intercellular colonization (*Ma et al., 2011*). The cellulase activity was restricted only to *Pseudomonas* sp. J10, *B. cereus* J156 and *P. xylanexedens* J155. Despite that *P. xylanexedens* J155 is the lowest in terms of IAA and ammonia production and only intermediate in term of P solubilization, it stimulates wheat seedling, root and shoot development. This finding is in-lane with a report showing that *Paenibacillus* significantly increased both dry and fresh weight of inoculated wheat (*Zhao et al., 2015*). Interestingly, *B. cereus* J156 strain promotes wheat germination, seedlings growth, shoot length, shoot dry weight and seedling vigor index, a finding in-line with other studies (*Raju et al., 1999*; *Raj et al., 2003*; *Raj, Shetty & Shetty, 2004*; *Kamran, Shazia & Shahida, 2010*), in which *Triticum aestivum's* seeds germination was increased following inoculation with either *Pseudomonas* or *Bacillus* strain (*Kamran, Shazia & Shahida, 2010*; *Bal et al., 2013*).

As the soil is a heterogeneous habitat and represents a broad spectrum of different ecological niches, it is well admitted that bacterial resistance to antibiotics facilitates their survival among the microbiome communities. Except for *B. megaterium* strain J11 and *B. cereus* J156, all other strains were resistant towards at least one of the six tested antibiotics. Soils are important reservoirs of diverse antibiotic resistance genes that can increase rapidly in clinical settings through horizontal gene transfer. Therefore, agricultural soils could play a major role in antibiotic resistance transmission. Additionally, a previous report has provided evidence for the exchanges of antibiotic resistance genes between soil (environmental) bacteria and clinical pathogens (*Ramakrishna, Yadav & Li, 2019*). To enable a sustainable agriculture and an effective antibiotic policy, it become necessary to unravel the conditions modulating the abundance of resistance genes in their microbial environment (*Kang et al., 2017*). Based on our results, and to avoid future use of bacteria conferring multi-resistance to antibiotics, the two *bacillus* strains *B. megaterium* J11 and *B. cereus* J156 that are sensitive to antibiotics, may represent potential safer use as biostimulants candidates.

Here, we also addressed the capacity of our strains to grow under increasing concentrations of heavy metals such as cadmium sulfate, nickel nitrate and copper sulfate. Pollution by heavy metals is considered as the main contaminants of our food supply, especially crop production (*Chauhan & Chauhan, 2014*). Contamination of plants occurs through the absorption of heavy metals from the soil, air, and water (*Sharma, Agrawal & Marshall, 2008*; *Singh et al., 2010*). Consequently, several adverse effects due to heavy metals pollution are considered as a great concern to public health, environmental health, and agricultural production (*Fergusson, 1990*; *Msaky & Calvet, 1990*; *Ma et al., 1994*; *Goyer, 1997*). Compared to literature (*Chauhan & Solanki, 2015*), our strains *Pseudomonas* sp. J10, *B. cereus* J156, and *Pseudomonas* sp. J153 tolerate higher concentration of heavy metals. The ability to adapt to heavy metal stress by developing various resistance mechanisms is partly mediated by bacteria production of intracellular metal binding proteins (*Hashem & Abed, 2002*). Both *Pseudomonas* sp. J10 and *B. cereus* J156 exhibit an adaptive response against nickel nitrate at 1500 $\mu$g mL$^{-1}$ whereas *Pseudomonas* sp. J153 tolerated up to 1500 $\mu$g mL$^{-1}$ of either copper sulfate or cadmium sulfate. Moreover, these values are remarkably high compared to 200 $\mu$g mL$^{-1}$ defined as the minimal inhibitory concentration of Cd and Ni in relation to previously isolated soil-bacteria (*Chauhan & Solanki, 2015*). However a recent study reported that *Bacillus cereus* tolerates up to 1500 $\mu$g mL$^{-1}$ of cadmium sulfate (*Khan et al., 2018*). Collectively, it appears clearly that *Pseudomonas* sp. J10 & J153 and *B. cereus* J156 strains possess the ability to withstand higher concentrations of heavy metals. Therefore, they are potential candidates suitable as bioremediatory agents for contaminated soils.

## CONCLUSIONS

The present study aimed to isolate and to characterize PSB isolated from the jujube plant (*Ziziphus lotus* (L.) Desf.). Among the nine tested strains, *Pseudomonas* sp. J12 and *Bacillus cereus* J156, emerged as potential bioinoculants as they share multiple beneficial

conventional PGP traits and promote seedlings wheat growth. In addition to *Bacillus cereus* J156, strains *Pseudomonas* sp. J10 and J153 emerged as potential candidates suitable to bioremediate heavy metals contaminated soils. To our knowledge, these results represent the first report on *Ziziphus lotus* (L.) Desf. associated PSB in Morocco and their role in promoting plant growth and alleviating heavy metals contamination. Here, we propose the use of the four promising strains *Pseudomonas* sp. J10, *P. moraviensis* J12, *Pseudomonas* sp. J153 and *B. cereus* J156, either for soil fertilization or heavy metals toxicity alleviation. Nevertheless, future agronomic studies on the field, using various plants, are required to explore elected bacterial strains' dual role in biofertilization and in the bioremediation processes.

## ACKNOWLEDGEMENTS

We thank Mohamed Hijri for helpful discussions and facilities.

### Funding
This work was supported by a welcome grant of the Mohammed VI polytechnic university (UM6P). Nidal Fahsi and Ismail Mahdi were supported by a PhD thesis grants from the UM6P. The funders had no role in study design, data collection and analysis, decision to publish, or preparation of the manuscript.

### Grant Disclosures
The following grant information was disclosed by the authors:
Mohammed VI polytechnic university (UM6P).
UM6P.

### Competing Interests
The authors declare there are no competing interests.

### Author Contributions
- Nidal Fahsi conceived and designed the experiments, performed the experiments, analyzed the data, prepared figures and/or tables, authored or reviewed drafts of the paper, and approved the final draft.
- Ismail Mahdi performed the experiments, analyzed the data, prepared figures and/or tables, and approved the final draft.
- Abdelhalem Mesfioui and Abdelmounaaim Allaoui conceived and designed the experiments, authored or reviewed drafts of the paper, and approved the final draft.
- Latefa Biskri conceived and designed the experiments, analyzed the data, authored or reviewed drafts of the paper, and approved the final draft.

### DNA Deposition
The following information was supplied regarding the deposition of DNA sequences:
16 S RNA gene sequencing: MT771625–MT771633.

## Data Availability

Raw data are available in a Supplemental File.

## Supplemental Information

Supplemental information for this article can be found online at http://dx.doi.org/10.7717/peerj.11583#supplemental-information.

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
