# Peer review of "Phosphate solubilizing rhizobacteria isolated from jujube ziziphus lotus plant stimulate wheat germination rate and seedlings growth"

_PeerJ, doi:10.7717/peerj.11583_

## Round 0.1 · original submission · Major Revisions

Dear Dr. Allaoui,

Regarding your MS - Phosphate solubilizing rhizobacteria isolated from Jujube ziziphus lotus plant stimulate wheat germination rate and seedlings growth - it is my conviction that it deserves major revisions before being acceptable for publication. In fact, although the MS is written in a clear and concise manner, it fails in highlighting the novelty of the work in face of similar studies. I would be glad to receive a new version, in which the points raised by the two reviewers are completely incorporated.

Sincerely,
Ana Ribeiro-Barros

Reviewer 1 ·

Basic reporting

The article is well structured and presented clearly. However, the text has some minor English errors that must be corrected. Please see "Comments for the author" for sugestions.
Literature references are generally adequate and suficient background/contex is provided.

Experimental design

The paper describes a solid and comprehensive research work that fits the Aims and Scope of the journal. Experimental procedures are adequate and carefully described.

Validity of the findings

The results are presented in an organized way and have been well analyzed. In general, conclusions are well stated but several points need to be clarified or modified. Some examples:
- It is not clear whether the indicated concentrations of heavy metals refer to the elements (Cu, Cd, Ni) or their salts.
- Informations about the experimental procedures are repeated in the Results section.Since they were described in detail in Materials & Methods, there is no need to include them in Results.
- Misleading sentences in Discussion: The data do not allow to conclude on the involvement of siderophores or ammonia production in the increases in shoots and roots length in the experimental conditions employed in the study. The production of IAA or other phytohormones was most probably involved in those effects.
Several other comments in "Comments for the author".

Additional comments

Line 21 – “jujube plant (Ziziphus lotus)”. The scientific name should be in italics.
Line 22 – “rhizospheric soil” or “rhizosphere”
Line 27-28 and throughout the text – Indole-3-acetic acid, in lower case
Line 31 – “the study of the potential of the isolates for promotion of early plant growth”
Lines 30-31 and throughout the text – Please clarify whether the indicated concentrations of heavy metals refer to the elements (Cu, Cd, Ni) or their salts.
Line 45 – “it is paramount”
Line 50 – “formulating more efficient”
Line 52 – Remove “but”
Line 55 – Replace “belongs” by “belong”
Line 60 – What do you mean by “phosphate solubilisation inorganic acids”? Please correct the sentence.
Line 64 – “are known as potent P solubilising microorganisms”
Line 70 – “(PGPR)”? Not clear why do you refer PGPR here.
Line 73 – “It is one of the major problems in deploying bacterial-based biofertilizers.” Are there experimental evidences for this, or is just speculation? Please provide a reference for this statement.
Line 86 - What do you mean by “inadvertently”?
Lines 96-98 – Sentence incomplete.
Line 102 – “jujube plants (Ziziphus lotus)”. The scientific name should be in italics.
Line 103 – “and, in vitro,”
Line 113 – Suspending, not dissolving
Line 133 – “Non-inoculated medium was used as blank”
Line 140 – Suggestion: “The experiments were performed in triplicate and the results are means of the replicates”
Line 153 and throughout the text – “16S rRNA gene” or “16S rDNA”
Line 153 – polymerase chain reactions (PCR)
L174 – Why 7 days? To ensure that the cultures had reached the stationary phase?
Line 197 – “Plates”
Line 221 – “sterile distilled water”
Line 225 – “were calculated using the following formula:”
Line 249 – “(20.5 mg.L-1)”
Line 254 – Genera, not genius
Line 266 – “tend”. Please give support for this statement.
Line 269 –Replace “confer” by “presented”
Line 274 – What about tetracycline?
Line 279 – I suggest to remove “trace elements” from this sentence.
Line 282 – Did you mean N2NiO6? (Also in Table 3)
Line 295 – Remove ;
Lines 299-303 – This information is already in Materials & Methods and there is no need to repeat it here.
Line 309 - This must be written in another way. Siderophores are best known for binding Fe and for mobilizing soil-immobilized Fe, although they may also contribute to improve the availability of P to plants.
Line 314 – CAS
Line 315 and throughout the text – “sp.” should be written in lowercase and not italic.
Line 320-321 – The Reference (Mahdi et al., 2020) is not adequate here, since it does not demonstrate (with results) what is stated in the sentence.
Line 321 – inhibitor
Line 322 – was
Line 321-323 - This information is already in Materials & Methods and there is no need to repeat it here.
Line 325 – “lowest one,”
Line 327-328 – Section title: “Proteases are not produced by Pseudomonas sp. J153 and J154 and cellulase activity is restricted to Pseudomonas sp. J10, J155 and B. cereus J156”
Line 340 – “… promote the highest rate of wheat seeds germination and seedlings growth”
Line 376 – “IAA is also frequently”
Line 377 – IAA is not a physiological precursor of auxin synthesis.
Line 386 – “…J13 was the…”
Line 386-390 – The sentence in line 387 is misleading. The data do not allow to conclude on the involvement of siderophores in the increases in shoots and roots length in the experimental conditions employed here. The production of IAA or other phytohormones was most probably involved in those effects.
Line 393-394 – The sentence is not clear, please re-write.
Line 397 – “trait”
Lines 397-400 – Same as above (see comment for lines 386-390).
Line 428 – Replace “trace elements” by “heavy metals”
Line 443-445 – The last part of the sentence (line 445) is not well written.
Line 448 – “jujube plant (Ziziphus lotus).”
Figures 1, 3, 4, 5 – Footnote: “Letters a, b and c highlight significant differences at p < 0.05”. You have more than these letters. Please correct or replace by: “Different letters indicate significant differences at…”
Table 1 – Please include the accession number of the closest sequence in GenBank or SILVA databases.
Table 2 – Footnote: The concentration of chloramphenicol is missing. The abbreviation for streptomycin (Sterp) is misspelling. There is no need to include the abbreviations of tetracycline and streptomycin since they are not mentioned in Table 2.

Reviewer 2 ·

Basic reporting

English: correct.
References: correct
Structure: correct
Relevant results: insufficient

Experimental design

Aims and scope: in line
Research question: unfortunately, nothing new!
Investigation performed rigorously with low technical standard. Ethical aspect fine
Methods well described

Validity of the findings

Not enough new elements to interest a large audience

Additional comments

This work on the PSBs is very interesting, but the publication of these intermediate results does not seem to me to meet international standards. Continue to develop inocula with these bacteria and work on longer plant growth times and be sure to highlight the functional role of your inoculates in the effects you will see on plants.

Annotated reviews are not available for download in order to protect the identity of reviewers who chose to remain anonymous.

---

## Round 0.2 · Minor Revisions

Dear Dr. Allaoui
Thank you for sending the revised version of the manuscript, which has been considerably improved. Nevertheless, before being accepted for publication, I suggest three minor changes:

- In the abstract, rather than providing a detailed report of part of the results, I suggest that you also highlight the novelty of this work based on the integrated data analysis: PGP traits, ARG, and bioremediation; if possible pointing out the most promising rhizobacteria.

- In the introduction: highlight the ecological importance of jujube

- conclusions, as suggested for the abstract, highlight the novelty of this work based on the integrated data analysis: PGP traits, ARG and bioremediation; if possible pointing out the most promising rhizobacteria.

Looking forward to receiving the new version,

Ana I. Ribeiro-Barros

Reviewer 1 ·

Basic reporting

English errors were corrected according to sugestions in the previous review.
However, commas are used several times throughout the text as a decimal separator, instead of periods. This should be corrected.

Experimental design

No comment.

Validity of the findings

The previous review drew attention to several aspects that were not clear. These have now been clarified or the text modified according to the suggestions.

Additional comments

The header of Table 2 indicates Table 1.

---

## Round 0.3 · accepted · Accept

Dear Dr. Allaoui,
Thank you for sending the revised version of the MS. I am glad to inform you that the paper is now accepted for publication. However, during the proofreading process, I suggest that you edit the text.

Thank you for choosing PeerJ,

Sincerely
Ana I. Ribeiro-Barros